# Improved Voltammetric Procedure for Chloride Determination in Highly Acidic Media

**DOI:** 10.3390/ma18010136

**Published:** 2024-12-31

**Authors:** Rafał Maciąg, Wojciech Hyk, Tomasz Ratajczyk, Mikołaj Donten

**Affiliations:** 1PIT-RADWAR S.A., Poligonowa 30, 04-051 Warsaw, Poland; 2Faculty of Chemistry, University of Warsaw, Pasteura 1, 02-093 Warsaw, Poland; wojhyk@chem.uw.edu.pl (W.H.); tk.ratajczyk@uw.edu.pl (T.R.); donten@chem.uw.edu.pl (M.D.)

**Keywords:** acid copper plating bath, chloride determination, voltammetry, electrodeposition, printed circuit boards

## Abstract

Cyclic voltammetry (CV) can be applied as a reliable method for the determination of chloride ions in a range from several to a couple hundred (about 200) ppm. Since the standard potential of chloride ion/gaseous chlorine is 1.36 V vs. normal hydrogen electrode (NHE), the efficient oxidation of Cl^−^ ion occurs at very positive electrode potentials, usually higher than +1.7 V vs. NHE. It is possible to observe this phenomenon only at noble-metal or inert carbon electrodes. Many other solutes, usually organic compounds, are often oxidized at this potential. This is the reason why the determination of Cl^−^ content directly from an increase in the oxidation current is not reliable and could lead to overestimated values. However, gaseous chlorine, generated at a positive potential dissolve in the analyzed solution, could be reduced in the reverse scan of a cyclic voltammetric curve. Optimization of the experimental procedure using statistical tools enables the development of an improved method for the direct quantification of chloride ions in acid copper electroplating baths. Under the proposed experimental conditions, the calibration curve (Cl_2_ voltammetric reduction current vs. chloride ions concentration) is represented by the linear model for the concentration of chlorides ranging from 10 to 200 mg/dm^3^. The developed method for analyzing chloride ions in an acid sulfate electroplating copper bath has many unique properties. It is fast; the time of a single analysis is less than 20 min. In automatic mode, it can be repeated up to 50 times a day. The method does not require processing of the sample of the analyzed bath before measurement. As a result, no additional chemical reagents are used, and the test sample can be returned to the plating bath.

## 1. Introduction

Currently, acid copper electroplating is a key technology in the semiconductor industry and the production of printed circuit boards (PCBs) [1,2,3]. In both applications, the copper electroplating process is a commonly used method of producing interconnects. The acid copper electroplating baths contain copper sulfate, sulfuric acid, chloride ions, and organic additives such as accelerators (grain refiners), suppressors, and leveling agents [4,5]. The use of organic additives is essential because their absence results in a copper deposit with properties that are significantly degraded (e.g., coarse-grained structure and low ductility). Therefore, mechanical properties like ductility and tensile strength are insufficient for industrial applications [6]. The presence of organic additives like suppressors, accelerators, levelers, and chloride ions in the acid copper plating bath change the cathodic copper ion reduction. It creates a synergistic effect with all the organic components [7]. Suppressors and chloride ions are absorbed at the surface of the cathode, which lower the plating rate [8,9]. As a result, the copper coatings are fine-grained with equiaxial grain structure. That guarantees a high ductility and tensile strength, which are critical features for high quality and reliability of PCBs [10].

In the damascene process [11], the presence of chloride ions within the described concentration range is essential to produce defect-free copper deposits (free from voids, seams, dimples, bumps). When consumable (copper) anodes are used, chloride ions aid anode corrosion, setting up a uniform and adherent anode film. The facts mentioned earlier, regarding the influence of chloride ions on the cathodic reduction of Cu^2+^ ions, demonstrate that achieving consistent and high-quality copper deposition under industrial conditions requires strict control of the chloride ion concentration within a very narrow range. The method of chloride analysis for use in the electronic industry should be characterized by the shortest possible time to enable the highest possible frequency.

The methods of chloride ion determination in aqueous solutions are usually based on volumetric [12], potentiometric—Ion Selective Electrode (ISE) [12,13], liquid chromatography [12,14,15], and spectrophotometry techniques [12]. All of them give reliable results in media with low or moderate ion strength (diluted solutions of accompanying agents) and preferably neutral pH. These methods are effective in the determination of Cl^−^ ions in water from various sources, such as sea water, tap water, and selected waste water [12]. The application of methods listed above are difficult or even impossible for samples containing a high concentration of other ionic substances, especially acids or complexation agents. In this case, analyzed samples must be specifically adapted to technical requirements of applied techniques (titration, potentiometric, or liquid chromatography) [13,16]. These specific requirements usually eliminate the aforementioned methods as a direct analytical method for the determination of chloride ions in electroplating baths. It is especially difficult in the case of analyzing acidic copper plating baths. The commonly used Mohr titration procedure [17] requires a pretreatment of the analyzed solution and often gives results significantly different from the real value. These reasons induced our efforts to develop a simple direct method for the determination of Cl^−^ ions in the range of 10 to 100 ppm in highly acidic media containing high concentrations of copper sulfate. The natural simple solution of the problem is applying a voltammetric method. Chloride ion—gaseous chorine is a redox couple with a standard potential of 1.36 V [18]. This means the signal of oxidation in the Cl^−^ ion should be noticeable in an acidic aqueous solution. The main issue covered by this paper is the method of chloride determination adopting linear scan and cyclic voltammetry (CV) [19] as reliable analytical procedures efficient in the determination of chlorine in acidic media of high ionic strength.

## 2. Materials and Methods

### 2.1. Apparatus

Electrochemical measurements and electrodeposition were carried out with the electrochemical workstation OrigaFlex-OGF01A (OrigaLys ElectroChem SAS, Lyon, France, OrigaMaster 5, Version: 2.1.0.3). The temperature of the measured bath was kept at 25 ± 1.0 °C.

CV was chosen as an electrochemical method for determining the content of chloride ions in acid copper sulfate plating baths used for the production of Printed Circuit Boards (PCBs).

The chloride determination process was carried out in a specially designed voltammetric cell. The designing, manufacturing, and testing of a measuring cell was performed in-house. A prototype of the voltammetric cell is illustrated in Figure 1. The working volume of the measuring cell is approximately 70 cm^3^ ± 10%. Commercially available completely inert plastic was selected as the building material for the production of a dedicated measuring cell. The reference electrode (RE) was prepared in this cell. In addition, this cell was also used to perform analytical measurements of the tested sample of the electroplating solution.

Cylindrical polycrystalline platinum electrode with a surface about 0.38 cm^2^ (dimensions—length 12 mm, diameter 1 mm) was employed as a working electrode (WE). The counter electrode (CE) was a polycrystalline platinum pipe-shaped foil, the surface of which was approximately 20 times larger than the surface of the working electrode. RE was a cylindrical polycrystalline platinum electrode with a surface about 0.38 cm^2^ covered with a copper layer (copper RE (Cu^2+^/Cu)). The copper film was produced by electrodeposition at a cathodic current density (Direct Current—DC) of 1.5 A/dm^2^. During the electrodeposition of a copper layer on the surface of a polycrystalline platinum electrode, efficient mechanical agitation of the plating bath was applied.

Cathode current density was controlled by an OrigaFlex-OGF01A galvanostat/potentiostat. Deposition time was fixed at 5 min, which resulted in a 1.3 ± 5% µm thick layer. Two anodes were used in the system. This option was used to obtain a copper layer of the most uniform thickness for the entire surface of the platinum cylindrical cathode.

All the potentials recorded in this study were related to this RE (Cu^2+^/Cu). For all CV measurements a three-electrode system was used. CV measurements of chloride content were performed between 0.6 V and 1.35 V at a scan rate of 0.1 V/s.

### 2.2. Analytical Measurement Procedure

A special analytical procedure has been developed for chlorides as a component of the acid copper sulfate plating baths. Quantitative measurements of chlorides were carried out using the CV technique.

The procedure for quantitative determination of chlorides consists of the following steps:Quasi-Reference Electrode (QRE) (Cu^2+^/Cu) PreparingPreparation of the WE surface in order to obtain constant and repeatable measurement conditions (electrochemical treatment by application of a potential of +1.8 V vs. Cu^2+^/Cu for 20 s)Quantitative measurement scan preparation (circulation period of the tested sample for 20 s, rest period of the tested sample for 20 s, application of 4 voltametric cycles in the range 0.6/1.35 V)Performing a quantitative measurement of chlorides (application of one voltametric cycle in the range 0.6/1.35 V vs. Cu^2+^/Cu at 0.1 V/s).Ending of the measuring sequence (electrochemical treatment by application of a potential of +1.8 V vs. Cu^2+^/Cu for 20 s)

### 2.3. Chemicals

The copper sulfate pentahydrate concentrate solution of semiconductor grade (Moses Lake Industries, Inc., Moses Lake, WA, USA) and H_2_SO_4_ (analytical reagent grade) were used for preparing the Virgin Makeup Solution (VMS), which is a plating solution without any added organics (i.e., zero accelerator, suppressor, leveler, organic and inorganic contamination). The additives used were ELECTROPOSIT™ M Additive (suppressor) and ELECTROPOSIT 1300 S from DuPont (Marlborough, MA, USA) formerly Dow (accelerator), and the chloride anions sourced from HCl (standard solution 1 mol/dm^3^, Chempur, Piekary Śląskie, Poland).

The ELECTRODEPOSITION^TM^ 1300 Acid Copper for PWB Metallization Applications plating process is designed for reliable through–hole plating of multilayer and double-sided PCBs. Deionized water (resistivity around 18 MΩ-cm, Millipore, MiliQ™, (Merck, Darmstadt, Germany)) was used in preparation of all solutions. The tested electrolyte composition was obtained from industry recommended values (*). The composition of the acid copper plating bath is presented in Table 1.

## 3. Results

### 3.1. RE Selection (Prerparation) and Testing Its Potential Stability

A three-electrode system was used in the development of the voltammetric method of chloride determination. The system requires an RE. The use of commercially available REs (silver chloride or saturated calomel electrode) in chloride determination in acid sulfate baths, used in PCB production, was proven unsuitable. The commercial electrodes may contaminate the analyte with chloride ions; additionally, the electrode needs topping up periodically.

The need to maintain the concentrations of all working components of the acid copper plating bath in a very narrow working range, including the concentration of copper sulfate (within ±5% of the optimal value), makes it possible to use a system consisting of a cylindrical platinum electrode covered with a copper layer as the QRE [20].

The copper film was produced by electrodeposition at a cathodic current density of 1.5 A/dm^2^. Two anodes were used in the system. This solution was used to obtain a copper layer of the most uniform thickness for the entire surface of the cylindrical cathode. The acid copper plating bath constituents for QRE and the plating parameters were given in Table 2.

Three deposition times were selected: 2 min, 5 min, and 15 min. The average thickness increased in proportion to the electrolysis time. For 2 min, the thickness was approximately 0.5 µm. However, for 5 min it was about 1.3 µm and for 15 min it was about 4 µm. Quasi-reference electrodes generated for 5 min were selected for Open Circuit Potential (OCP) tests.

The QRE (Cu^2+^/Cu) potential stability was quantified using open circuit cell potential [21] measured against silver chloride electrode (with saturated KCl solution) at 298 K (25 °C).

Three types of copper plating solutions were tested. The first one was a VMS, that was an acid copper plating bath containing optimal concentrations of inorganic components and free from any organic or inorganic contamination. The second solution type was a newly prepared acid copper plating bath containing optimal concentrations of organic and inorganic components and free from any organic or non-organic contamination. For the third type of solution, a sample was taken from the production plating bath after aging increased by 150 Ah/L. Chemical composition of the sample of the acid copper bath used for the OCP stability measurements are presented in Table 3.

This selection of the acid copper plating bath sample used in industrial conditions resulted from the fact that with the process of electroplating, the accumulation of a series of by-products of plating solution additives will directly affect the hole filling ability and deep plating ability of the plating solution and the uniformity of copper coating thickness. Improper growth control process will directly lead to loose, rough, porous, or broken copper coating surface crystals, resulting in excessive hole filling depression value, insufficient deep plating, and decreased uniformity of copper coating thickness [4].

To conduct studies on the stability of the QRE potential using the OCP method, the following procedure was proposed. In the first stage of the study, a QRE was made for each bath sample (described in Table 3), and then its stability was tested in a freshly prepared plating bath (describe in Table 3 as the newly prepared acid copper plating bath). In the second stage, the QRE was re-manufactured for each tested sample of the plating bath (except for the VMS sample). This time, the OCP method measurement was conducted in the same solution from which the QRE was made.

Since the obtained data are very similar to each other, regardless of the stage of life of the electroplating bath, for a better presentation of the obtained data, it was decided to present the results obtained for the bath at two diametrically different stages of its life in the form of a graph. Figure 2 and Figure 3 present examples of test results for the stability of the QRE (Cu^2+^/Cu) potential generated from the acidic copper plating bath at various stages of the bath’s life. The rest of the obtained data showing the stability of the QRE potential was presented in numerical form in Table 4.

Data obtained from OCP measurements indicate that the condition of acid copper plating solution contaminated due to aging of bath did not affect the stability of the potential quasi-reference electrodes.

The results clearly indicate the potential raise over initial 80–100 s, which then remains stable until the end of the measurement period (1 h). The initial change in potential happens due to adsorption–desorption processes occurring on the surface until equilibrium was reached. During the process of cathodic reduction of copper (II) ions, their concentration (oxidized form) directly at the cathode surface is lower than in the bulk. After interrupting the electrolysis process, the system needs time to re-establish equilibrium, which, according to the Nerst equation, must lead to an increase in the recorded potential.

For the OCP measurements performed, the average recorded potential was 92 mV. These data are collected in Table 4.

The QRE (Cu^2+^/Cu) stability studies have demonstrated its suitability. The electrode delivers a stable potential over a period required for chloride determination (approximately 5 min). Additionally, each measurement could be performed using a freshly prepared quasi-reference electrode.

Figure 4 is showing the relation between electrical potential, various concentrations of the copper in the ELECTROPOSIT™1300 Acid Copper bath and temperature of the solution.

The working range for copper concentration in the ELECTROPOSIT™1300 Acid Copper bath by DuPont was from 0.18 to 0.22 mol/dm^3^. During the experiments, the temperature was maintained between 296 and 300 K. Assuming an average temperature of 297 K, the potential could be calculated using the Nernst equation (for a concentration of 0.2 mol/dm^3^ of Cu (II)_aq_, after conversion to activities). The calculated potential was 95 mV.

The small difference between the theoretical value and the actual formal potential determined from OCP measurements may result from the assumptions made when calculating activity coefficients. This approach was necessary because the ionic strength of the examined galvanic bath was approximately 3.

### 3.2. Characterization of the Method

The method presented in the article bases on oxidation and reduction signal of Cl_2_/Cl—redox couple. Others reported in the literature procedures were indirect ones and based on interactions of chloride ions with other systems such as the oxidation and reduction of silver [22], ferrocenemethanol [23], or Co(II)-based anion receptors [24]. None of the methods can be used in highly acidic media such as an acidic copper plating bath.

The technique proposed in the paper is based on the direct oxidation signal of the Cl^−^ ion and it was designed for an acidic copper sulfate bath, but it can be utilized in other media, especially highly acidic.

The proposed method of determination of Cl^−^ concentration was based on the electrochemical behavior of redox couple chlorine/chloride ion. The standard potential of this system in aqueous media was determined as 1.36 V vs. NHE and did not depend on the pH of the solution. The electrochemical window of the Pt electrode covers the potential in acidic solution [25,26]. Thus, the voltammetric method could be applied at low pH of the analyte. Also, the method was not affected by high concentration of many ions which could not be oxidized or reduced at similar to Cl_2_/Cl^−^ redox couple potentials. This was a substantial advantage of the procedure because other methods, such as titration, ion chromatography, ion selective potentiometry, and even coulometry are strongly affected by pH and ion strength of the analyte. So, the listed methods could not be directly applied, for example, in analysis of acid copper electroplating baths. In the developed procedure, chloride ion concentration could be determined directly from oxidation current of Cl^−^, observed at very positive potentials, or reduction current of gaseous chlorine, generated during oxidation of Cl^−^ ions. Both signals were potentially useful for analysis. At the potential range from 0.4 V to oxygen evolution, no other significant current signals could be observed.

Figure 5 displays cyclic voltammograms for chloride oxidation/chlorine reduction. These cyclic voltammograms were recorded for a freshly prepared (unused) acid copper plating electrolyte (containing organic additives at optimal concentration levels). During the initial forward scan (from 0.6 to 1.35 V), an increasingly oxidation potential was applied.

Intensity (current) of both of the signals depends on concentration of chloride ions present in the analyzed solution. The linearity of the oxidation current is compared to the concentration dependence apparent; however, the signal could be disturbed by the oxidation current of organic additives and contaminants of the plating solution. Thus, the oxidation signal could be used only in uncontaminated solutions and was not very reliable in production baths. In such a situation, the analysis could be performed by measuring the reduction current of gaseous chlorine in the reverse half-cycle. Stabilization of this signal can be performed by repeating the oxidation reduction cycle up to obtaining steady-state conditions. In our experimental case, cycling was performed in a potential range of 0.60 to 1.35 V, with scan rate of 0.1 V/s. The 5th cycle was taken for measurement.

Analyses of chloride ion/chlorine redox couple voltammogram indicate that the system is reversible in meaning the existence of electroactive reduced and oxidized forms in the solutions. However, separation of oxidation and reduction peaks suggests that not transport of the electroactive species, but the kinetics of the electrode reactions is what limits the redox process. Moreover, the kinetic limit of the current does not interfere with the linearity of the correlation plot current vs. chloride ion concertation.

### 3.3. Calibration of the Method

The calibration plot was carried out for solutions identical with acid copper plating solution used for PCB fabrication (composition reported in Table 1). Figure 6 displays calibration plot of peak current vs. Cl^−^ concentration.

In the proposed method of Cl^−^ ion determination, the top limit of its suitability was determined by solubility of gaseous Cl_2_ in the analyzed solution. According to the literature data, the limit of chlorine concentration in the analyzed bath was circa 48 mmol/dm^3^ [27,28]. Above this limit, the calibration plot should reach a plateau. At the reported experimental conditions, the calibration plot was linear up to concentration of Cl^−^ of 200 mg/dm^3^, which is compatible with the operational range of the chloride ion concentration in an acid copper electroplating bath. Modification of the experimental condition, particularly the scan rate, should shift the range of linearity of calibration plot to higher or lower values.

Calibration was a crucial stage in developing an analytical method for the quantification of chloride ions in acidic media. It results in the construction of a linear relationship between directly measured voltammetric signals—the limiting currents of the Cl_2_ reduction process (I_red_)—and the corresponding chloride concentrations (c_Cl_) in the prepared matrix samples spiked with a known amount of chloride ions standard. Since both correlated variables were affected by various measurement errors, their experimental values were associated with standard uncertainties. In the case of the dependent variable (voltammetric current), the standard uncertainty (u(I_red_)) was estimated using the A method [29], i.e., based on the magnitude of the random dispersion (repeatability) of the measurement results (four replicated independent voltammetric measurements). In the case of an independent variable (chloride concentration), the estimated standard uncertainties (u(c_Cl_)) were combinations of several independent sources of uncertainty resulting from the adopted sample preparation scheme, i.e., dilution of the certified reference material (CRM–-hydrochloric acid). At least three sources of uncertainty could be distinguished: concentration of the CRM analyte, volume of the CRM sample (pipette limiting error), and the total volume of the system (volumetric flask limiting error). The combination of these uncertainty sources yields the combined relative uncertainty in the chloride concentration at the level of 1%.

Table 5 presents a complete list of the numeric data employed for the construction of the calibration curve. In the calculation process a linear relationship I_red_ = a·c_Cl_^−^ + b, where a and b were the slope and the intercept of the straight line was assumed. To obtain the most representative estimates of the values of coefficients a and b and their standard uncertainties (u(a) and u(b)), it was required to employ a general weighted linear regression scheme for calculations, taking the uncertainties of both scales into account and the significant heterogeneity of uncertainties in the measured values of the dependent variable. In the weighted regression calculation scheme the extent to which a given experimental point determines the course of the calibration line was quantified by its weight coefficient. Uncertainties of both correlated variables contribute to the weight coefficient in the following form (Equation (1)):W_i_ = [u^2^(I_red,i_) + a^2^u^2^ (c_Cl_^−^_,i_)]^−1^(1)
where i numbers indicate the experimental points employed for the regression calculation.

The calculated values of weights are listed in Table 5.

Under the proposed experimental conditions (described in the previous section), the obtained regression model for the concentration dependence of the voltammetric current can be represented by the following equation (Equation (2)):I_red_ = (−0.834 ± 0.018) c_Cl_^−^ − (0.78 ± 0.34)(2)
with the correlation coefficient *r* = −0.9998.

It was found that the linear dependence between the Cl_2_ reduction current and the chloride ions concentration was valid for the concentration of chlorides ranging from 10 to 200 mg/dm^−3^. The upper level of the proposed concentration range was considerably above the typical Cl^−^ concentration found in electroplating baths. It was also important to note that any change in the voltammetric conditions (e.g., scan rate, the type of RE etc.) requires recalibration of the developed analytical method.

A calibration curve is also a convenient tool for the estimation of the method quantification limit (LOQ) according to the following formula (Equation (3)):LOQ = p × u(b)/a(3)
where u(b) is the standard uncertainty of the intercept and p represents a multiplication constant (usually set to 10).

The LOQ parameter for the proposed method was found to be 2 ppm.

The proposed method was used for the determination of chloride concentration in copper plating bath used for fabrication of PCBs. In this case, there is no simple direct method for the determination of Cl^−^ level in the bath solution. Our effort was turned to determine this component in a range usually expected in the copper plating bath solution, which is 10 to 100 ppm (mg/dm^3^) [30,31]. Commonly applied argentometric titration is a time-consuming process and requires modification of the sample [13]; thus, it cannot be considered a procedure for continuous monitoring of the chloride ions in the copper plating bath. Additionally, such a procedure generates a significant amount of waste which has to be treated. Our voltammetric method requires collection only of a small portion of the bath solution, performing the analytical procedure, and returning the analyzed portion back into the monitored plating bath. Such proceeding could be applied because the sample of the solution used for analysis was not modified during the determination process.

The newly developed electrochemical method of chloride ion determination (based on cyclic voltammetry) in acid copper plating bath was compared to a titration method, commonly used for analysis of copper plating baths.

The determination of Cl^−^ ion concentration was performed for two solutions. The first one was a newly prepared bath sample of recommended composition shown in Table 6. The bath was prepared using 0.1 M hydrochloric acid standard solution as a chloride source, which was transferred using automatic micropipette into a 1 dm^3^ A-class volumetric flask. The second sample was collected from the plating bath which was working for a significantly long time. The bath was operating properly but was not strictly controlled for the concentration of the constituents, especially organic additives.

The comparison results of analyses performed by voltammetric and titration techniques are shown in Table 7. The voltammetrically determined Cl^−^ ion contents indicate significant differences compared to the modified argentometric titration results. However, the reliability of the data were confirmed by additional analyses performed for intentionally spiked bath solutions. For both of them, the concentration of chlorides was increased two times, by two consecutive spikes, each of 25 mg/dm^3^ (ppm). The proportional response observed for the implemented method proves its reliability and shows, beyond doubt, that it can be applied not only in the form of comparison result with calibration plot but also using standard addition method (single or multiple). It means that the developed chloride ion determination method may be directly used for working plating line (continuous monitoring) and for electrolyte samples (analyses performed in offline mode), as well.

## 4. Discussion

The described voltammetric method of determination of chloride ion content in aqueous solutions may be an attractive alternative for commonly used methods such as application of ISEs and argentometric titration. The advantage of voltammetric determination of concentration of Cl^−^ ion is its good performance in solutions of high ionic strength and high concentration of acid. In such experimental conditions, usage of ISE is impossible and titration is not reliable due to problems with detection of the titration end point. Thus, the presented method could be applied in determination of concentration of chloride ion in acidic copper plating baths.

The analytical procedure was fit to conditions and the range of concentration of analyte in commercially used electrolytes, i.e., 10–200 ppm, however the range of determined concentration may be changed by modification of experimental parameters, e.g., scan rate. The performed investigation indicated that the method gives reliable results in newly prepared and heavily used solutions. An additional advantage of the method is possibility of its automation integration with industrial copper plating process. It was found that the method was also stable for off-line analyses based on standard addition (simple or multiple). In summary, the proposed analytical method is comparable to other methods commonly used in analysis of chloride ion and additionally gives good results in aggressive media such as acid copper plating baths.

## 5. Conclusions

The difficulty of accurate determination of chloride concentration in highly acidic media was successfully solved applying a voltammetry technique. The described method is simple, fast, and reliable which was proved applying it for Cl^−^ analysis in an acid sulfate copper plating bath. A complete analytical system containing stable and robust RE-based on Cu (II)/Cu redox couple and inert Pt WE was proposed and proven for this analytical technique. The constructed system does not require any maintenance even for months of operation. Since the sample solution is not modified during analytical procedures, it can be returned to plating bath. Thus, the method does not generate chemical waste as titration (up to now the only one efficient way to determine Cl^−^ level in highly acidic media). Additionally, the described method is easy to integrate with other electrochemical procedures used for plating bath monitoring and is fast (less than 20 min for whole cycle). In automatic mode it could be repeated up to 50 times a day. The method is extremely stabile and in appropriate cleaning and conditioning Pt working electrode, e.g., by oxidation during vigorous oxygen evolution, repeatable results can be obtained through couple of months. The appropriate tests support the statement.

## Figures and Tables

**Figure 1 materials-18-00136-f001:**
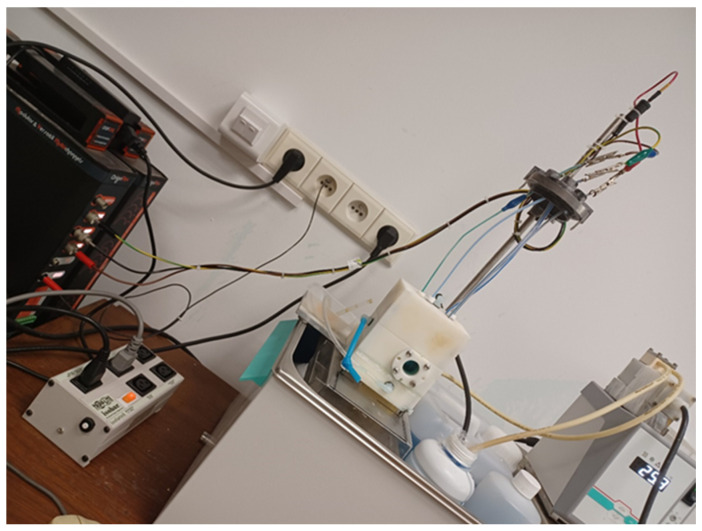
A prototype of the voltammetric cell.

**Figure 2 materials-18-00136-f002:**
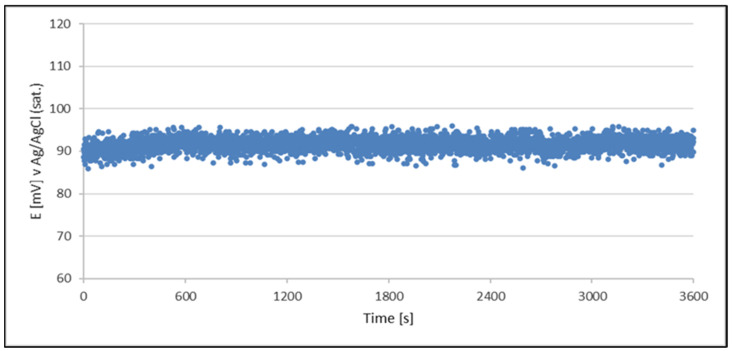
OCP variation with time for a QRE (Cu^2+^/Cu) made from a sample of a newly prepared acid copper plating bath The QRE was tested in a newly prepared acid copper plating bath.

**Figure 3 materials-18-00136-f003:**
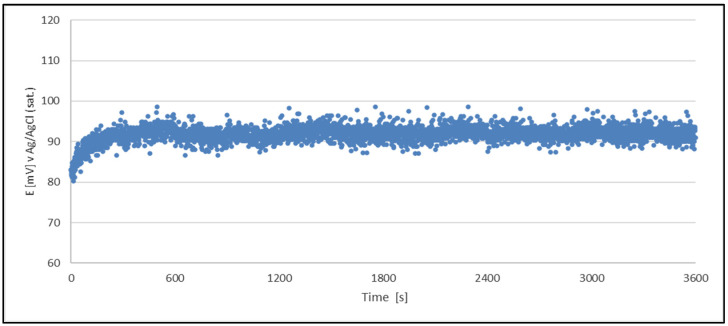
OCP variation with time for a QRE (Cu^2+^/Cu) made from a sample of a production acid copper plating bath. The QRE was tested in the production acid copper plating bath too.

**Figure 4 materials-18-00136-f004:**
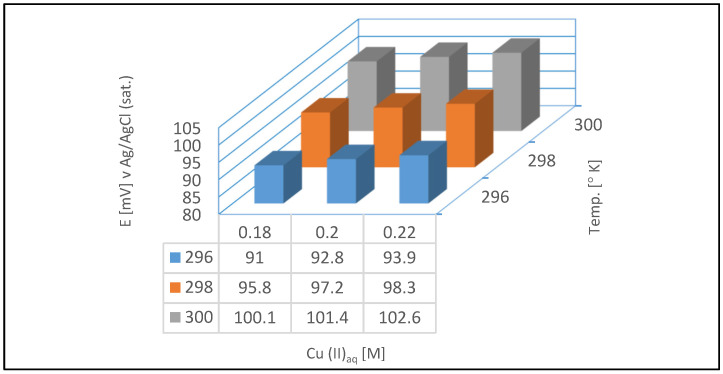
Potential changes depend on variations in copper concentration and temperature, which were permissible within the operating range for DuPont’s ELECTROPOSIT™1300 Acid Copper technology.

**Figure 5 materials-18-00136-f005:**
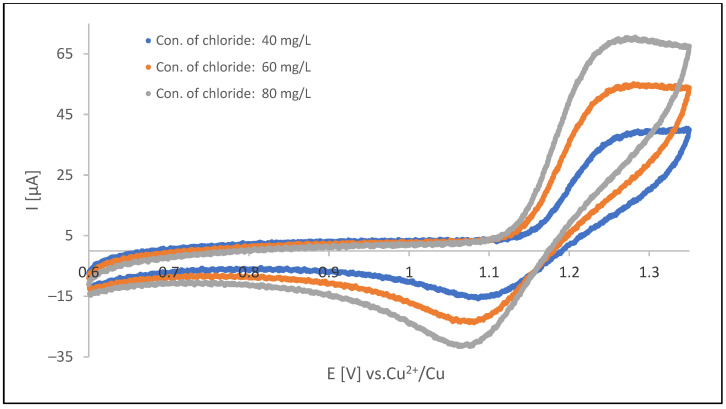
Cyclic voltammograms recorded at acid copper plating bath. Signals at about 1.28 V correspond to Cl^−^ oxidation signals at about 1.08 V corresponds to Cl_2_ reduction. All three CV curves were registered at sweep rates of 0.1 V/s for each direction of potential sweep. These voltammograms were recorded at a polycrystalline Pt electrode (temperature 298 K).

**Figure 6 materials-18-00136-f006:**
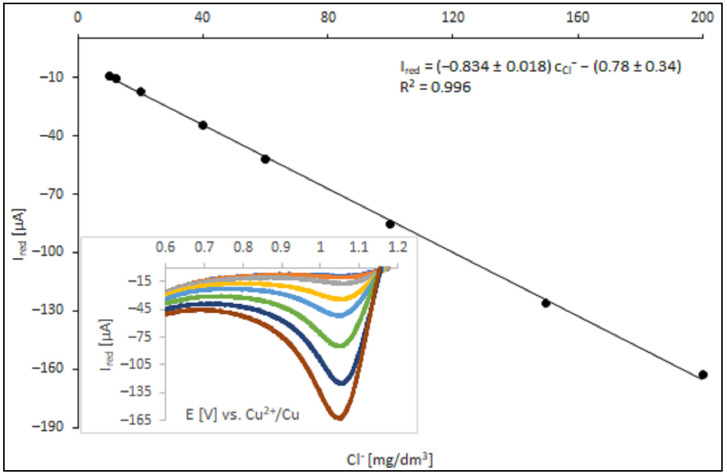
The calibration curve for chloride concentration ranging from 10 to 200 mg/dm^3^. The colors of the curves correspond to the following chloride ion concentrations: dark blue—10 mg/dm^3^; orange—12 mg/dm^3^; gray—20 mg/dm^3^; yellow—40 mg/dm^3^; blue—60 mg/dm^3^; green—100 mg/dm^3^; navy blue—150 mg/dm^3^, brown—200 mg/dm^3^.

**Table 1 materials-18-00136-t001:** Chemical composition of the tested copper plating bath.

Component *	Range *
Copper (II) (as CuSO_4_·5H_2_O)	0.18–0.22 M (45.0–55.0 g/dm^3^)
Sulfuric Acid	2.45–2.65 M (240.0–260.0 g/dm^3^)
Chlorides	1.13–1.69 mM (40.0–60.0 mg/dm^3^)
ELECTROPOSIT™ M Additive	40 cm^3^/dm^3^
ELECTROPOSIT 1300 S	1 cm^3^/dm^3^

(*) Technical data sheet, Electrodeposition ^TM^ 1300 Acid Copper for PWB Metallization Applications.

**Table 2 materials-18-00136-t002:** Chemical composition and plating parameters of the copper acid sulfate bath called ELECTRODEPOSITION^TM^ 1300 Acid Copper for PWB Metallization Applications by DuPont, used for the production of electrical contacts in PCBs.

Component	Range
Copper (II) (as CuSO_4_·5H_2_O)	0.18–0.22 M (45.0–55.0 g/dm^3^)
Sulfuric Acid	2.45–2.65 M (240.0–260.0 g/dm^3^)
Chlorides	1.13–1.69 mM (40.0–60.0 mg/dm^3^)
ELECTROPOSIT™ M Additive	40 cm^3^/dm^3^
ELECTROPOSIT 1300 S	1 cm^3^/dm^3^
Temperature	298 K (25 °C)
Cathodic Current Density (DC)	1.7 A/dm^2^

**Table 3 materials-18-00136-t003:** Chemical composition of the samples of the copper acid sulfate bath called ELECTRODEPOSITION^TM^ 1300 Acid Copper (For PWB Metallization Applications) by DuPont, used for the OCP stability measurements.

	Concentration of Components of the Acid Copper Plating Baths Used for OCP Measurements		
Components of the Acid Copper Plating Bath	The Virgin Makeup Solution	The Newly Prepared Acid Copper Plating Bath	The Production Acid Copper Plating Bath
CuSO_4_·5H_2_O	0.20 M (51.0 g/dm^3^)	0.20 M (51.0 g/dm^3^)	0.21 M (52.4 g/dm^3^)
H_2_SO_4_	2.60 M (255.0 g/dm^3^)	2.55 M (255.0 g/dm^3^)	2.55 M (250.0 g/dm^3^)
Cl^−^	1.41 mM (50 mg/dm^3^)	1.41 mM (50 mg/dm^3^)	1.30 mM (46 mg/dm^3^)
ELECTROPOSIT™ M Additive	0 cm^3^/dm^3^	Unknown **	Unknown **
ELECTROPOSIT™ 1300 S	0 cm^3^/dm^3^	Unknown **	Unknown **

(**)—the content of organic additives was monitored by Hull Cell analysis (plating time: 10 min, DC amperage: 1.5, temperature: 298 K (25 °C), mixing bath: air agitation, polished brass test plate).

**Table 4 materials-18-00136-t004:** Recorded range of potential changes during OCP measurements for the produced quasi reference electrodes.

Type of Acid Copper Bath Used for Preparing Quasi-Reference Electrode	OCP Measurements for a Newly Prepared Acid Copper Plating Bath Average Value [mV]	OCP Measurements for a Production Acid Copper Plating Bath Average Value [mV]
The Virgin Makeup Solution	92.0	N/A ^a^
The newly prepared acid copper plating bath	91.6	N/A ^a^
The production acid copper plating bath	92.6	91.8

a—N/A it means that there is no data.

**Table 5 materials-18-00136-t005:** Experimental results for the construction of the calibration curve.

No.	c_Cl_^−^ [mg/dm^3^]	u(c_Cl_^−^) [mg/dm^3^]	Ired [µA]	u(Ired) [µA]	Weight, W
1	10	0.10	−9.10	0.058	97.3
2	12	0.12	−10.733	0.088	56.2
3	20	0.20	−17.50	0.32	7.63
4	40	0.40	−34.66	0.47	3.04
5	60	0.60	−51.85	0.55	1.82
6	100	1.0	−85.13	0.93	0.644
7	150	1.5	−126.9	1.1	0.355
8	200	2.0	−163.53	0.19	0.355

**Table 6 materials-18-00136-t006:** Composition of copper plating solutions utilized for verifying the new method of chloride determination.

	Concentration of Components of the Acid Copper Plating Baths	
Components of the Acid Copper Plating Bath	A Newly Prepared Acid Copper Plating Bath	A Production Acid Copper Plating Bath
CuSO_4_·5H_2_O	0.21 M (52.3 g/dm^3^)	0.22 M (54.9 g/dm^3^) ***
H_2_SO_4_	2.60 M (255.0 g/dm^3^)	2.55 M (250.0 g/dm^3^) ***
Cl^−^	1.41 mM (50 mg/dm^3^)	1.11 mM (39 mg/dm^3^) ***
ELECTROPOSIT™ M Additive	40 cm^3^/dm^3^	unknown
ELECTROPOSIT™ 1300 S	1 cm^3^/dm^3^	unknown

(***) values determined in commonly used analytical methods (titrations).

**Table 7 materials-18-00136-t007:** Chloride content for two different acid copper plating baths obtained by two different analytical methods.

C	Chloride Concentrations in the Tested Acid Copper Plating Baths						
	Parameters			Results			
	P1 [mg/dm^3^]	P2 [mg/dm^3^]	P3 [mg/dm^3^]	R1 [mg/dm^3^]	R2 [mg/dm^3^]	R3 [mg/dm^3^]	R4 [mg/dm^3^]
**A**	50	+25	+25	55	77	105	32
**B**	unknown	+25	+25	24	47	76	39

Where C (a type of the tested acid copper plating baths), A (a newly prepared acid copper bath), B (a production acid copper plating bath), P1 (assumed initial conc.), P2 (first spike), P3 (second spike), R1 (initial conc. determined by newly developed analytical voltammetric method), R2 (results of the first spike), R3 (results of the second spike), R4 (initial concentration determined by standard method (modified argentometric titration)).

## Data Availability

The original contributions presented in the study are included in the article; further inquiries can be directed to the corresponding author.

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
