# Peer review of "Improved Voltammetric Procedure for Chloride Determination in Highly Acidic Media"

_materials, 2024, doi:10.3390/ma18010136_

Round 1

Reviewer 1 Report

Comments and Suggestions for Authors

Optimization of the experimental 18 procedure using the statistical tools enables the development of the improved method for the direct quantification of chloride ions in acid copper electroplating baths.

1. The abbreviation that appears for the first time requires writing all the original words (such as NHE in the abstract).

2. Suggest expanding the introduction section as the content is insufficient.

3. The angle taken in the Figure 1 is clearly a bit off, it is recommended to take the picture positive.

4. It is recommended to use a three-line table for the table (such as table 1-7). 

5. There are a few references, so it is necessary to cite more recent ones.

Author Response

Dear Sear or Madam,

A PDF file containing the responses to the suggestions is attached.

Best regards,

Rafał Maciąg

Reviewer 2 Report

Comments and Suggestions for Authors

Comments: Authors report the improved voltammetric procedure for chloride determination in highly acidic media. The structure of the synthesized materials has been characterized well, and these analyses are reasonable. However, the authors should address the following comments to ensure its acceptance.

1.   Some significant numerical results are missing in the abstract. Hence, the abstract needs to be improved with some numerical results drawn from the studies to attract the reader's attention.

2.    How is the synthesis method/work different or better than those reported earlier? Author should highlight the novelty and the importance of the present work in the introduction part.

3.    I suggest merging Figures 2 - 5 for a clear understanding of readers.

4.    What about the stability process?

5.    In order to show the superiority of the current materials, comparisons over the other related materials reported in the past literatures are necessary. Detecting the performance of the current materials has to be compared with that of the other materials, and reasons for performance improvements have to be discussed.

6.    There are some grammatical errors in the manuscript which have to be rectified for clear understanding.

Comments on the Quality of English Language

Minor spell checks are required. 

Author Response

Dear Sir or Madamm,

A PDF file containing the responses to the suggestions is attached.

Best regards,

Rafał Maciąg

Reviewer 3 Report

Comments and Suggestions for Authors

Maciag et al. materials-3303868

In this manuscript, the authors report an improved method for cloride concentration determination in highly acidic media using cyclic voltammerty. The reulsts show linear relationship between chloride concetration and current within the common range of commercial electrolytes.

Comments:

(1)     Please check the uses of plurals vs singlulars through the manuscript.

(2)     It might be better to combine Figure. 2 to Figure. 5 into a signle figure for better present the data to the audience.

(3)     Define open circuit potential (OCP) at the first place it appears. For example, on line 195 it is defined. However, it appeared several times before the definition.

(4)     Figure 6 is not mentioned or refered in the text.

(5)     Please elaborate more on the reversibilities of the Cl2/Cl- redox couple from the CV data shown in Fig. 7.

(6)     Give R2 value for fig. 8.

(7)     Check spells through the text such as on line 127, it should be “sulfate” not “sulphate”.

In summary, this manuscirpt is recommended for publication in materials after minor revision.

Author Response

Dear Sir or Madam,

 A PDF file containing the responses to the suggestions is attached.

Best regards, 

Rafał Maciąg
